# A Perceived Dissociation Between Systemic Chronic Inflammation, Age, and the Telomere/Telomerase System in Type 2 Diabetes

**DOI:** 10.3390/biomedicines13030531

**Published:** 2025-02-20

**Authors:** Mai S. Sater, Dhuha M. B. AlDehaini, Zainab H. A. Malalla, Muhalab E. Ali, Hayder A. Giha

**Affiliations:** 1Department of Biochemistry, College of Medicine and Health Sciences (CMHS), Arabian Gulf University (AGU), Manama P.O. Box 26671, Bahrain; dhuha.aldaihani@hotmail.com (D.M.B.A.); zainabhm@agu.edu.bh (Z.H.A.M.); muhalabae@agu.edu.bh (M.E.A.); 2Kuwait Medical Genetic Center, Al-Assima (Kuwait) P.O. Box 22488, Alsafat, Kuwait City 13085, Kuwait; 3Medical Biochemistry and Molecular Biology, Khartoum, Sudan; gehaha2002@yahoo.com

**Keywords:** T2D, inflammation, age, telomeres biology, inflammatory markers, genotyping

## Abstract

**Background**: Chronic inflammation is associated with leukocyte telomere length (LTL) shortening and type 2 diabetes (T2D). The latter is also associated with LTL shortening, while the three variables are associated with aging. **Objective:** It is tempting to test whether inflammation, age, or both are behind the telomere system aberrations in diabetic patients. **Methods**: In this cross-sectional observational study, blood samples collected from 118 T2D patients were analyzed via ELISA to estimate the plasma levels of four inflammatory markers, IL6, IL8, TREM1, and uPAR, and the telomerase enzyme (TE). Moreover, the extracted DNA was used for the LTL estimation via qPCR and for single nucleotide polymorphisms (SNP) genotyping of TE genes (TERT, TERC, and ACYP2) via rtPCR. **Results:** The results showed no correlation between the levels of all tested inflammatory markers and the LTL, TE level, and age. There were no significant differences between the marker levels in diabetic patients in the four quartiles of the LTL and TE levels. Moreover, there were no significant differences in the levels of the markers between carriers of the different TE genotypes. **Conclusions:** There were no associations between the tested inflammatory markers’ levels and the LTL, TE plasma levels, or age in T2D. Explanations for the dissociation between the above-known associations in T2D were proposed; however, the subject is worth further investigation.

## 1. Introduction

Systemic chronic inflammation (SCI) is a low-grade, persistent process classically triggered by damage-associated molecular patterns DAMPs [1], aging [2,3], and chronic disorders [4]. Importantly, SCI causes damage to vital organs and plays a role in the progression of chronic diseases over time, such as metabolic syndrome [5,6] and type 2 diabetes (T2D) [5]. Diabetic chronic inflammation is partially attributed to gut dysbiosis (alteration of gut microbiota) [7], apoptosis, chronic metabolic acidosis, oxidative stress, and others [3,8]. Currently, there are no standard biomarkers for chronic inflammation generally or taler-designed for T2D. However, it has been shown that canonical biomarkers of acute inflammation can predict morbidity and SCI [9], but with notable limitations. Inflammation is a known cardinal feature of T2D, which is marked by raised inflammatory biomarkers including the classic markers, e.g., IL-6, TNF-α, IL-1β, cytokines, and CRP, and the non-traditional ones, e.g., TREM-1 and uPAR [10].

The sources of inflammatory molecules in SCI are senescence-associated secretory phenotype (SASP) cells [11], which are senescent cells that over-express proinflammatory cytokines, chemokines, and other pro-inflammatory molecules [11]. Both endogenous and exogenous factors can induce an SASP. The endogenous factors are DNA damage, oxidative stress, and dysfunctional telomeres [12]. The environmental contributors include obesity [13], microbiome dysbiosis [14], and diet [15], which are all associated with T2D.

Telomeres are nucleoproteins, composed of repetitive nucleotide sequences (TTAGGG)n combined with peptides, that guard the chromosomal ends against damage by exonucleases, free radicles, and other noxious factors [16]. Hasty unprogrammed telomere shortening may lead to chromosomal recombination and fusion and neoplasia. Apart from cellular division, telomeric erosion can be exaggerated by extrinsic factors such as oxidative stress and inflammation [17]. The increased risk of adverse health outcomes and aging attributed to telomere shortening is documented in several studies [18,19,20]. On the other hand, the implication of SCI in telomere attrition in normal aging has been demonstrated in previous studies [9,21].

The extension of telomeres is accomplished by the telomerase enzyme (TE). The TE is expressed in germ line tissue and omnipotent stem cells. However, it is only basally expressed in most somatic tissue; therefore, there is steady shortening of telomeric repeats. The TE is also a ribonucleoprotein complex. It is composed of two major subunits, the catalytic subunit, which is the reverse transcriptase (TERT), and the telomerase RNA component (TERC), which serves as a template for the synthesis of the telomeric repeats. Other proteins are required for TERT/TERC core complex stabilization, e.g., dyskerin, NHP2, NOP10, and GAR1 [22]. In addition to its biological role in telomere maintenance in highly proliferative cells (germ, stem, and embryonic cells), the TE has a pathological role, as it is critical for most tumor cells [18]. The insufficient amount of TE in adult somatic cells is probably behind the failure of eroded telomere repair during aging [18,23]. Technically, the leucocyte telomere length (LTL) is used as a representative of the overall bodily chromosomal telomere length.

This study aims to investigate the possible role of inflammation in the telomere’s attrition and TE aberrations in T2D by testing the correlations of LTL, TE levels, and gene polymorphism with the levels of the inflammation biomarkers IL-6, IL-8, TREM-1, and uPAR and the IL6/IL8 ratio, with the consideration of age as a confounding factor. Since we do not know the nature of inflammation and its relevant markers in T2D or the telomere system, the choice of diverse biomarkers broadens the scope of the test and increases the chance of detecting inflammation. Therefore, we selected cytokines (IL6), and chemokines (IL8), the classical and most commonly tested inflammatory markers, and also non-interleukins (TREM1, and uPAR), the non-classical and less widely tested ones. It is worth noting that the same set of markers was previously shown to be associated with T2D in the same setting and was also tested in obesity and sepsis, where it was shown to discriminate between the study groups.

## 2. Materials and Methods

### 2.1. Study Area and Subjects

This cross-sectional study aimed to investigate the possible role of chronic inflammation in T2D on telomere biology by correlating and comparing variables of T2D, inflammation markers, and telomere system components. The study was carried out in Kuwait in 2018, as part of a project about the telomeres system in T2D. All study subjects were original Kuwaiti natives; the patients (118 subjects) were recruited from the diabetic clinics of hospitals and primary health centers of the Ministry of Health in Kuwait, while the controls (74 non-diabetic subjects) were volunteers recruited via the electronic media. Patients with T2D were not seriously ill and did not have chronic disorders other than the usual complications of diabetes. Patients with cancer or chronic inflammatory disorder or those on long-term anti-inflammatory drugs were excluded. The controls were seemingly healthy.

### 2.2. Data Collection

Data were collected from T2D patients and non-diabetic healthy volunteers by obtaining their personal, demographic, and clinical (including disease, drug, and family histories) information, followed by clinical examination.

### 2.3. Ethical Issues

Ethical approvals were obtained from the Research and Ethics Committees of the Kuwait Ministry of Health (2015/242) and the College of Medicine Medical Sciences, Arabian Gulf University (E28-PI-01/20). Informed consent was obtained from each participant.

### 2.4. Blood Sample Collection

Overnight fasting (10 to 12 h) blood samples were collected from the study subjects into EDTA-containing and plain vacutainers, then the samples were thoroughly mixed and centrifuged at 3000 rpm for 10 min. The samples were separated into plasma, buffy coat, and serum and stored at −20 °C or −80 °C in labeled cryotubes for subsequent analysis.

### 2.5. Clinical Chemistry Analysis

The clinical chemistry workup was performed in the hospitals’ chemistry labs. A UniCel1 DxC Synchron 800 analyzer (Beckman Corporation, Brea, CA, USA) and a TOSOH G8 High-Performance Liquid Chromatography Analyzer (TOSOH Bioscience, South San Francisco, CA, USA) were used for measurement of fasting blood glucose (FBG) and glycated hemoglobin (HbA1c) and total cholesterol (TC), triacylglycerol (TAG), high-density lipoprotein cholesterol (HDLC), and low-density lipoprotein cholesterol (LDLC), respectively. The Homeostatic Model Assessment for Insulin Resistance index (HOMA-IR) was calculated using the following formula: fasting insulin (micro-U/L) × fasting glucose (nmol/L)/22.5. 

### 2.6. Enzyme-Linked Immunosorbent Assay (ELISA): Plasma Insulin, Cytokines, and Telomerase Measurement

#### 2.6.1. Insulin

The plasma insulin level was measured using a solid-phase sandwich ELISA kit from Invitrogen (KAQ1251), as shown in the protocol provided. The detection limit was 0.17 µIU/mL.

#### 2.6.2. Inflammatory Markers

The levels of inflammatory markers were estimated by solid-phase sandwich ELISA using Invitrogen ELISA kits, EH2IL6 (for IL-6), KHC0081 (for IL-8), EHTREM1 (for TREM-1), and EHPLAUR (for uPAR), following the protocols provided with the kits, as mentioned before [10]. A Thermo Multiscan Spectrum Plate Reader coupled with SkanIt RE for MSS 2.4.2 software was used for measuring the plates’ absorbances.

#### 2.6.3. Telomerase

The plasma TE was measured using a quantitative sandwich ELISA kit from MyBioSource Inc. (San Diego, CA 92195-3308, USA), MBS021959, by following the manufacturer’s instructions [24]. Microtiter plates coated with captured antibodies directed specifically against TE as an antigen were used for the analysis. The plasma samples contained the antigen (TE), and the controls were titrated into the wells and incubated. Following the antigen–antibody interaction, the plates were washed and then detection antibodies were added and incubated, a step followed by another wash. Thereafter, a conjugated horseradish peroxidase (HRP) was added and the mixture incubated. The wells were washed again before the addition of the substrate solution. An ELISA reader at 450 nm was used for reading the optical density (OD) within 15 min of adding the stop solution. The plasma concentration of TE was calculated using the standard curve. The kit sensitivity was 1.0 U/L and the detection range was (3.12 U/L–100 U/L).

### 2.7. DNA Extraction

A QIAamp^®^ DNA blood Mini Kit was used for genomic DNA extraction from whole blood and buffy coats, and a Nanodrop spectrophotometer was used to test the concentrations of the extracted DNA samples and the A260/A280 ratio.

### 2.8. Estimation of the Absolute Leukocyte Telomere Length via Real-Time PCR

The Absolute Human Telomere Length Quantification qPCR Assay Kit (AHTLQ–#8918) from ScienCell’s was used to estimate the mean LTL, while the Applied biosystem 7500 real-time PCR system was used for the run, as reported before [24]. A Single Copy Reference (SCR) primer set was used as a reference for data normalization, where it recognizes and amplifies a 100 bp long region on human chromosome 17. A reference genomic DNA sample with known telomere length (TL) was used as a reference for the calculation of the sample TL. Two qPCR reactions were prepared for each DNA sample, including the reference DNA, one with the telomere primer and one with the SCR primer stock solutions. Two replicates were performed for each sample. All the reactions were prepared in 96-well qPCR reaction plates.

### 2.9. Selection of TE Genes SNPs and Real-Time PCR (rtPCR) Genotyping

Three TE genes and SNPs, TERC rs12696304 (G/C), TERT rs2736100 (CA), and ACYP2 rs6713088 (GC), were selected based on their reported association with the telomere’s length. The genotyping was performed using the allelic discrimination method with the Applied Biosystem (ABI) StepOne Plus and 7500 real-time PCR systems according to the manufacturer’s instructions, as reported before [25]. The two principal steps are cycling (PCR amplification) and the endpoint detection of fluorescent signals. Selective annealing via TaqMan^®^ MGB probes was used for allelic discrimination.

### 2.10. Statistical Analysis

The statistical analysis was carried out using Sigma-Stat software (Systat Software Inc., version 3.5. Copyright 2006). The T-test, Mann–Whitney Rank Sum Test (MW), ANOVA, and Kruskal–Wallis One Way Analysis of Variance on Ranks (KW) were used in comparisons, according to the distribution and number of tested variables. The Pearson Product Moment Correlation was used in the correlation analysis. The statistical significance was set at *p* < 0.05.

## 3. Results

### 3.1. Characteristics of the Study Subjects

As seen in (Table 1), the study included 118 patients with T2D (53 males, 53 females, and 12 not reported) and 74 non-diabetic, apparently healthy Kuwaitis (27 males and 47 females). The age, HOMA-IR, plasma insulin, HbA1c, and lipid profile were significantly different between the two groups; however, their BMI values were comparable.

### 3.2. Correlations Between Age, LTL, and TE in T2D and Non-Diabetic Healthy Subjects

In T2D patients (Figure 1A), there was no correlation between age as a major confounding factor and LTL (CC −0.0751, *p* = 0.455) or TE concentration (CC −0.0178, *p* = 0.869); there was also no correlation between LTL and TE concentration (CC 0.112, *p* = 0.285). In the seemingly healthy subjects used as controls (Figure 1B), there was also no correlation between age and LTL (CC 0.141, *p* = 0.254) or between LTL and TE (CC −0.011, *p* = 0.931). However, there was a markedly significant correlation between age and TE concentration (CC 0.324, *p* = 0.005) via Pearson Product Moment Correlation.

### 3.3. Correlations of the LTL with the Plasma Levels of the Tested Inflammatory Markers

As seen in Figure 2A-i–E-i, only the plasma level of IL8 was correlated positively with the LTL, CC 0.239, *p* = 0.024, while the plasma levels of IL6, TREM-1, and uPAR and the IL6/IL8 ratio were not correlated with the LTL: CC −0.008, *p* = 0.939; CC, 0.231, *p* = 0.09; CC 0.175, *p* = 0.12; and CC 0.0375, *p* = 0.730, respectively, Pearson Product Moment Correlation.

### 3.4. Correlations of the Plasma TE with the Plasma Levels of the Tested Inflammatory Markers

The plasma TE (Figure 2A-ii–E-ii) was not correlated with the levels of all tested inflammatory markers: IL6, IL8, TREM1, uPAR, and IL6/IL8 ratio (CC −0.046, *p* = 0.686; CC 0.044, *p* = 0.695; CC −0.069, *p* = 0.622; CC −0.080, *p* = 0.498; and CC −0.021, *p* = 0.854, respectively, Pearson Product Moment Correlation).

### 3.5. Inter-Quartile Comparisons of the Plasma Levels of Inflammatory Markers in T2D Patients in the Different LTL and TE Quartiles

As seen in (Table 2), the levels of IL6, IL8, TREM1, and uPAR, as well as the IL6/IL8 ratio, in the LTL quartiles of T2D patients were not significantly different. Similarly, the levels of the four markers and ratio were comparable in the TE concentration quartiles of T2D patients.

### 3.6. Comparisons of the Plasma Levels of the Tested Inflammatory Markers Between Diabetic Patients Carrying Different Genotypes of the Telomerase Genes’ SNPs

As seen in Table 3, the plasma levels of IL6 were comparable between T2D patients carrying the genotypes of TE genes: *TERC* rs12696304 G/C (GG, CC, and GC), *p* = 0.463; *TERT* rs2736100 C/A (CC, AA, and CA), *p* = 0.974; and *ACYP2* rs6713088 G/C (GG, CC, and GC), *p* = 0.778. Similarly, the plasma levels of IL8 were comparable between the same TE genotypes, *p* = 0.179, *p* = 0.057, *p* = 0.200, respectively, as well as the TREM1 levels, *p* = 0.060, *p* = 0.552, and *p* = 0.841, respectively, and uPAR levels, *p* = 0.741, *p* = 0.782, and *p* = 0.210, respectively. Also, the IL6/IL8 ratio was not significantly different between the genotypes of the TE genes: *p* = 0.473, *p* = 0.880, and *p* = 0.392, respectively.

### 3.7. Correlations of the LTL and Plasma TE Concentration with the Plasma Levels of the Tested Inflammatory Markers in Healthy Subjects

As seen in the scatter matrix of Figure 3, only the plasma level of IL6 was correlated positively with the LTL, CC 0.316, *p* = 0.009, while the plasma levels of IL8, TREM-1, and uPAR and the IL6/IL8 ratio were not correlated with LTL in non-diabetic healthy subjects: CC 0.024, *p* = 0.846; CC −0.104, *p* = 0.401; CC 0.051, *p* = 0.682; and CC 0.196, *p* = 0.112, respectively,

The plasma TE level was not correlated with the levels of all tested inflammatory markers in non-diabetic healthy subjects, i.e., IL6, IL8, TREM1, uPAR, and the IL6/IL8 ratio: CC −0.004, *p* = 0.976; CC −0.095, *p* = 0.423; CC 0.105, *p* = 0.373; CC −0.08, *p* = 0.498; and CC 0.040, *p* = 0.732, respectively, Pearson Product Moment Correlation (Figure 3).

## 4. Discussion

Type 2 diabetes, inflammation, and LTL shortening are closely related to the aging process. Building upon this background, we aimed to investigate whether inflammation, age, or both are at the root of telomere disorders in T2D. Given the growing diabetes mellitus epidemic and population aging, understanding the mechanisms linking these processes is crucial for developing preventive and therapeutic strategies. Systemic low-grade chronic inflammation (SCI) is a cardinal feature of T2D, while aging is known to be associated with LTL shortening. In this setting, we previously reported the association of T2D with SCI [10] and LTL shortening [24] independently. In the current study, we used the same material and SCI biomarkers, but we failed to detect any causal relationship between inflammation and telomere/telomerase system abbreviations in T2D. Moreover, age, as a dependent or independent variable, was not associated with the telomere system or inflammation in diabetics. In contrast, it was correlated with TE level in healthy subjects.

Although inflammation and immune responses are responsible for both cytokine production and telomeric erosion, unexpectedly, the tested inflammatory markers, IL6, IL8, TREM1, and uPAR, were not correlated with LTL shortening, the TE level, or polymorphisms in this study. The exception to this was the levels of IL-8, which were correlated with LTL. In other studies, IL6 was shown to be associated with decreased LTL [21,26,27] and, controversially, it was shown to increase TE activity in vitro; it may, therefore, maintain telomere length [28,29]. However, another group of studies linked telomere/telomerase system aberrations to immune system dysfunction that leads to chronic inflammation [20,30]. Whether inflammation triggers telomerase/telomere dysfunction, or vice versa, remains to be elucidated [19,31]. These studies were not conducted in T2D and are not limited to relatively older aged patients (>40 Yrs.), as in this study.

All tested markers, except IL8, were previously shown to be significantly raised in T2D [10]. In this study, the IL8 levels were positively correlated with the LTL, i.e., high IL8 levels were associated with increased LTL, in contradiction to the role of inflammation in trimming the telomeres. Our results might indicate that SCI is not associated with LTL shortening in T2D, at least in this setting. The IL-8 produced by macrophages acts as a chemo-attractant for neutrophils during acute inflammation and induces the quick mobilization leukocyte precursors hematopoietic stem cells (HSCs). Therefore, IL8 is more likely to be an acute rather than a chronic inflammation biomarker. However, it was previously shown that the attributes of IL-8 might have important implications in telomere erosion since increased mitotic division is required to replenish mobilized HSCs [32]. Conversely, the IL8 levels in the present study were positively correlated with the LTL in T2D patients. Although the correlation was weak, it was significant and did not support the telomere attrition.

Moreover, the telomere/telomerase system gene polymorphisms did not affect the inflammatory marker levels in the present study. The genotype carriers of the tested SNPs, *TERC* rs12696304 G/C, *TERT* rs2736100 C/A, and *ACYP2* rs6713088 G/C, all had comparable levels of IL6, IL8, TREM1, and uPAR and similar IL6/IL8 ratios between the genotypes of each SNP. On the other hand, the *TERT* and *ACYP2* but not the *TERC* gene polymorphisms were previously shown to be associated with blood glucose levels and TE levels; however, they were not associated with LTL in T2D in this setting [25]. In another study, TERC was shown to promote the cellular inflammatory response independent of the complete TE [33].

Concerning the relationship of uPAR and TREM1 with the telomere system, one study showed that uPA (the ligand for uPAR) is positively associated with the TERT component of TE and that hTERT enhances uPA expression in cancer cells [34]. Other than this, we found no article on this topic. To our knowledge, no study has investigated the relationship between the TE and uPAR as a marker of chronic inflammation. Similarly, the association between TREM1 as a pro-inflammatory marker and the telomere system was not reported. Thus, we demonstrated in this study for the first time that both markers are not associated with the LTL, TE level, or TE gene polymorphisms in T2D.

SCI induces telomere dysfunction by increasing ROS-mediated DNA damage and thus accelerates the accumulation of senescent cells, limits tissue regeneration, and hastens aging [35]. In T2D, LTL was found to be correlated inversely with patients’ oxidative stress status [36]. Interestingly, this study showed that T2D buries the effects of inflammation and possibly age on the telomeres system, though it is known that T2D contributes to telomere attrition via ROS overproduction. It has also been observed that telomere loss in T2D patients contributes to oxidative stress and endoplasmic reticulum stress and that telomere shortening can serve as an independent risk factor for T2D, contributing to the disease progression [37]. Furthermore, telomere shortening in T2D involves mitochondrial dysfunction as an intermediate process that can lead to oxidative stress [38]. It is worth noting that oxidative stress is strongly linked to SCI.

Regarding the role of age in T2D, SCI, and the telomere system, it was reported that TL is inversely associated with circulating endotoxin levels independent of T2D, suggesting that SCI may trigger premature biological aging [35]. Age, the fundamental contributor to SCI and telomere attrition in healthy subjects [30], was not correlated with LTL or the TE level in diabetic subjects in this setting. This controversy can somewhat be explained by the fact that the study subjects have a narrow age range (52.0–64.0 years, 25–75% percentile) and a relatively advanced median age (56.5 years), while the influence of age on TL is more apparent in a full age spectrum that includes the young population. The negligible role of age on the LTL in this study is strengthened by the observation that even in the healthy non-diabetic subjects there was no correlation between age and LTL. Although they were significantly younger than the diabetic patients, the healthy subjects were mostly above fifty years of age, where the contribution of age to telomere attrition is expected to decline [39].

Interestingly, while the TE level was not correlated with age in T2D, it was in apparently healthy controls, which fosters a possible hypothesis that T2D overshadows the effect of age on the telomere system, including the TE. TE activity drops to the minimum level in most somatic cells in healthy adults [40], except for a few types of cells, e.g., germ line, stem, and fast-proliferating cells. On the other hand, TE activity is associated with healthy aging and decreased degenerative diseases via maintaining the telomeres that protect the cellular genome and increase cell lifespan and longevity [41]. Moreover, lifestyle changes, e.g., regular exercise and healthy food, have also been correlated with a modest increase in TE activity in immune cells [42]. Taken together, this could explain the increased TE activity in healthy subjects associated with increasing age in the present study, supporting the well-being of the control group. In contrast, T2D was shown to be associated with decreased TE activity [43], which was not recognized in this study. However, in certain pathological conditions, like malignancies, up to 90% of patients express TE significantly [44], while aging is a risk for cancer, emphasizing the role of aging in the TE profile.

The limitations in this study might be contributing to unraveling a latent role for SCI in telomere system aberrations in T2D. These include: (i) The used biomarkers not covering the different aspects of the inflammatory process in both T2D and the telomere system, like oxidative stress, chronic acidosis, nitric oxide, etc. (ii) The cross-sectional design might limit the ability to identify dynamic relationships between inflammation and LTL and track changes with time. (iii) If we speculate that SCI and LTL shortening exists only in certain subgroups of patients, here, the heterogenicity of T2D, stage of the disease, complications, or genetic factors will matter and need to be investigated. (iv) The measured plasma TE may not reflect the activity of the TE in leukocytes. Other factors affecting LTL such as age, genetics, lifestyle (diet, physical activity, smoking, stress), etc., need to be available. Although they will not change the result, they can help to explain the findings. Finally, our data do not completely disprove the link between inflammation and telomeres in T2D but rather indicate the need to revise existing perceptions and search for more complex and subtle mechanisms of interaction. Importantly, this study’s results and discussion emphasize the importance of negative findings as positive ones. Therefore, future lines of research on this subject should consider the T2D duration and genetics and heterogeneity as important variables in longitudinal research using a broader list of biomarkers in a larger number of study subjects in different ethnic groups.

In conclusion, the present study unveiled the lack of association between chronic inflammation and telomere biology, specifically telomere length; telomerase enzyme level; and TE gene polymorphisms in T2D. Also, the study uncovered the lack of influence of age on the telomere system in T2D. Alternatively, the global metabolic alternations in T2D may mask or override the effects of SCI and age on the telomere system. Finally, larger studies using different inflammatory markers in other settings are needed to confirm this study’s findings.

## Figures and Tables

**Figure 1 biomedicines-13-00531-f001:**
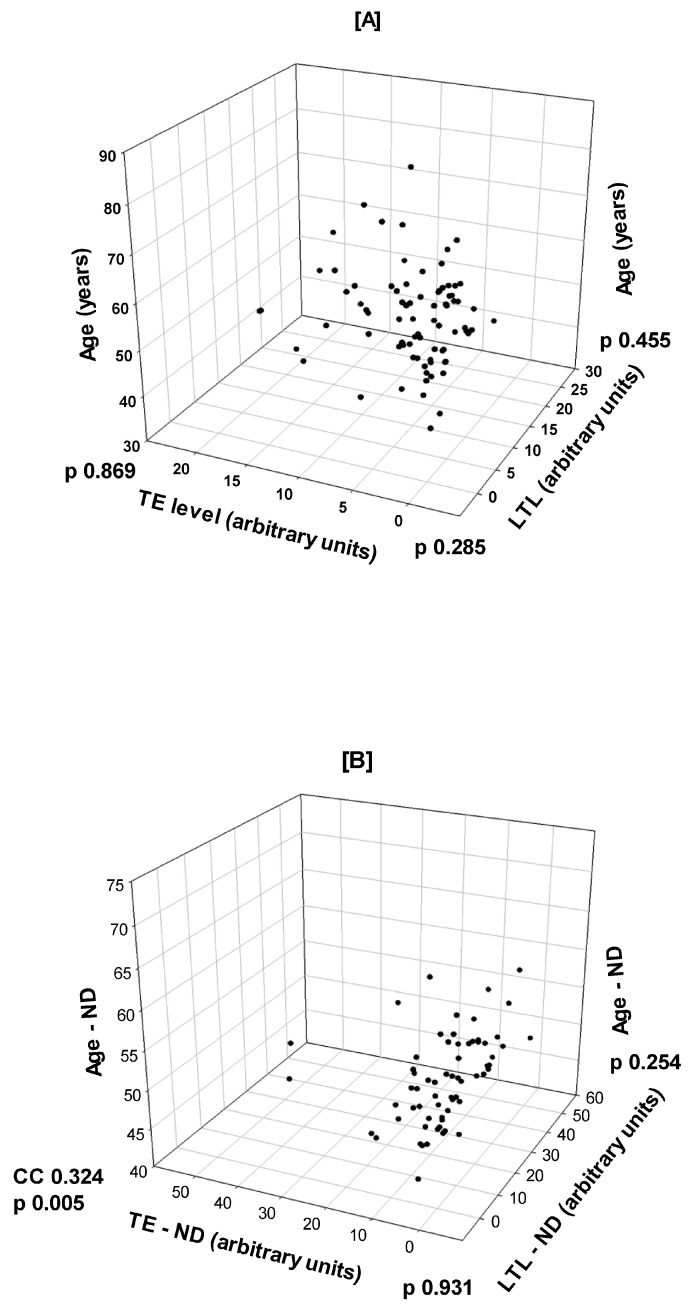
A three-dimensional plot showing: (**A**) a lack of correlation between age and leukocyte telomere length (LTL) (*p* = 0.455) and age and telomerase enzyme (TE) plasma level (*p* = 0.869). Also, there was no correlation between LTL and TE (0.285) in type 2 diabetes (T2D). (**B**) In apparently healthy non-diabetic subjects (ND), although there was no correlation between age and LTL (*p* = 0.254) and between LTL and TE (*p* = 0.931), there was a significant correlation between age and TE level (CC 0.324, *p* = 0.005) via Pearson Product Moment Correlation.

**Figure 2 biomedicines-13-00531-f002:**
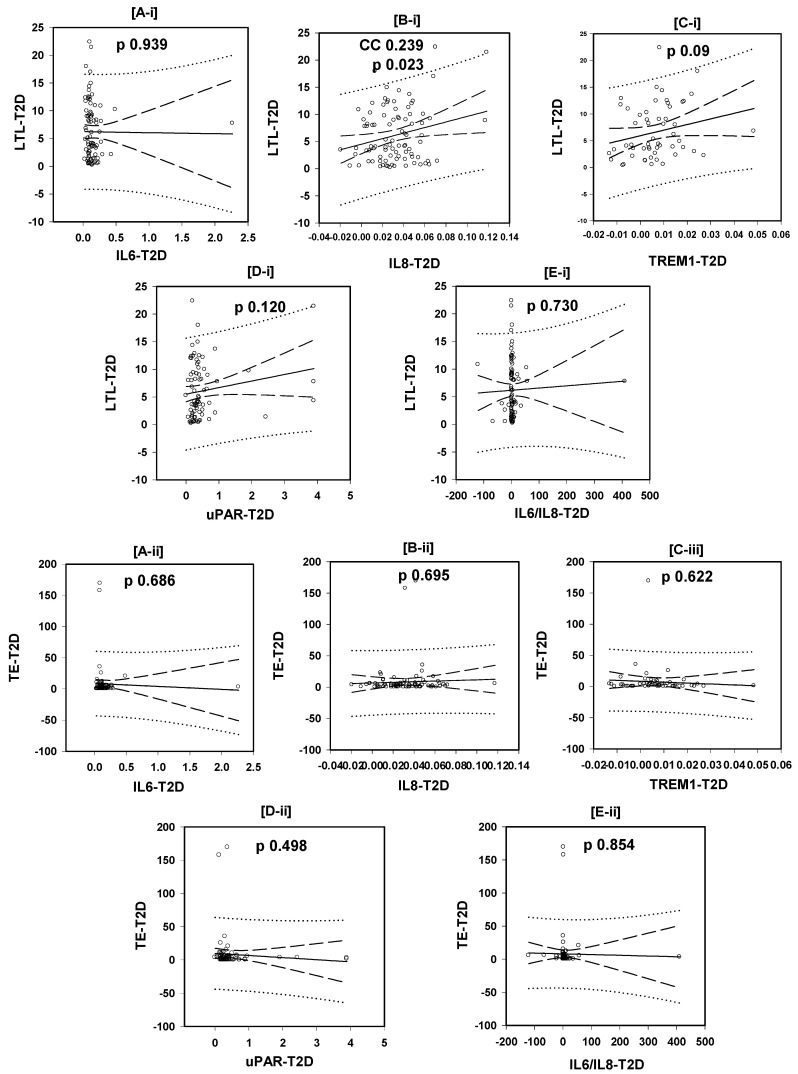
Demonstration of the lack of correlations between the LTL and inflammatory marker levels: IL6 (*p* = 0.939), TREM1 (*p* = 0.09), uPAR (*p* = 0.120), and IL6/IL8 ratio (0.730) (**A-i**, **C-i**, **D-i**, and **E-i**, respectively) in Kuwaiti T2D patients. The exception was (**B-i**), which shows the significant positive correlation between LTL and IL8 (CC 0.244, *p* = 0.023). (**A-ii**–**E-ii**) Demonstrate the lack of correlations between the TE and the inflammatory markers IL6 (*p* = 0.686), IL8 (0.695), TREM1 (*p* = 0.622), and uPAR (*p* = 0.498) and the IL6/IL8 ratio (*p* = 0.854) seen in the figures (**A-ii**–**E-ii**), respectively, in the same study subjects.

**Figure 3 biomedicines-13-00531-f003:**
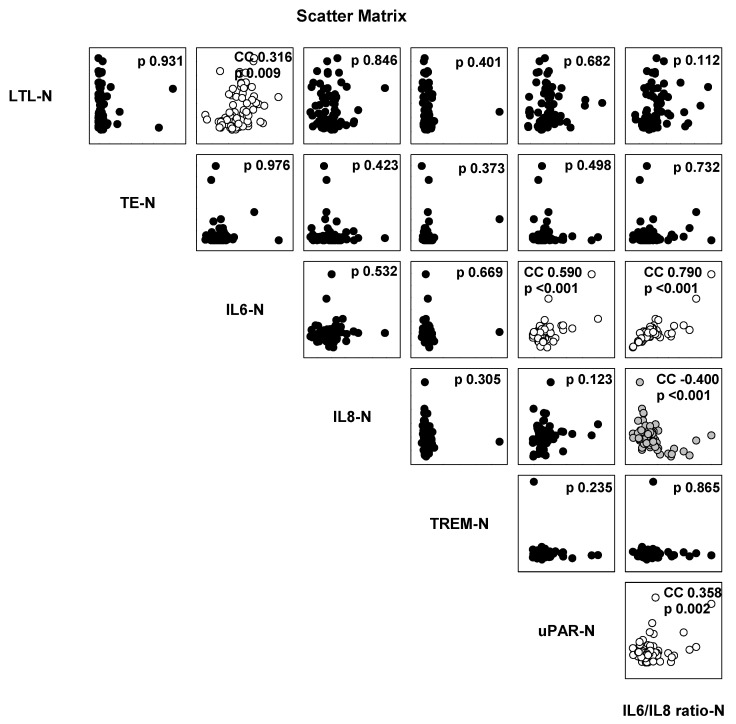
A scatter matrix plot showing the correlations between the telomeres’ system parameters and the inflammatory markers. LTL versus IL6 (CC 0.316, *p* = 0.009), IL8 (*p* = 0.846), TREM1 (*p* = 0.401), uPAR (*p* = 0.682), and the IL6/IL8 ratio (*p* = 0.112) and the TE level versus IL6 (*p* = 0.976), IL8 (*p* = 0.423), TREM1 (*p* = 0.373), uPAR (*p* = 0.498), and the IL6/IL8 ratio (*p* = 0.732) in apparently healthy non-diabetic subjects (N). The plots also show the correlations between the different parameters and the others. Open circles represent statistically significant correlations.

**Table 1 biomedicines-13-00531-t001:** Description of the study subjects and their biochemical profiles.

Variables	T2D Patients	Healthy Controls	*p*-Value	Stat Test
Number	118	74		
Sex (M/F)	53/53 *	27/47		
Age	56.5, 52.0–64.0	51.0, 49.0–55.000	<0.001	MW
BMI	30.81, 26.83–35.59	28.89, 26.56–33.30	0.149	MW
HOMA-IR	12.32, 7.85–17.60	2.76, 1.30–3.99	<0.001	MW
Insulin	34.09, 25.78–50.38	12.23, 8.72–16.37	<0.001	MW
HbA1c	7.65, 7.00–8.60	5.90, 5.65–6.25	<0.001	MW

MW: Mann–Whitney Rank Sum Test. * The sex of 12 patients was not reported.

**Table 2 biomedicines-13-00531-t002:** The plasma levels of the tested inflammatory markers in the LTL and TE concentration quartiles in T2D patients.

Inflammatory Marker	Levels (Arbitrary Units)				*p*-Value
LTL	1st Quartile	2nd Quartile	3rd Quartile	4th Quartile	KW
IL6	0.137, 0.092–0.189	0.139, 0.101–0.163	0.123, 0.081–0.160	0.110, 0.092–0.129	0.486
IL8	0.033 ± 0.022	0.023 ± 0.019	0.036 ± 0.025	0.038 ± 0.027	0.169 *
TREM1	0.0001 ± 0.0072	0.0058 ± 0.0110	0.0112 ± 0.0132	0.0089 ± 0.010	0.100 *
uPAR	0.258, 0.198–0.409	0.353, 0.280–0.428	0.399, 0.226–0.568	0.356, 0.209–0.399	0.329
IL6/IL8	3.635, 2.493–6.918	3.797, 2.143–9.022	3.828, 1.679–9.008	2.733, 1.832–4.842	0.672
TE Levels	1st Quartile	2nd Quartile	3rd Quartile	4th Quartile	
IL6	0.100, 0.075–0.166	0.143, 0.102–0.203	0.123, 0.104–0.173	0.100, 0.086–0.125	0.067
IL8	0.0336 ± 0.0176	0.0313 ± 0.0215	0.0285 ± 0.0298	0.0355 ± 0.0171	0.765 *
TREM1	0.0064 ± 0.0111	0.0093 ± 0.0157	0.0052 ±0.0098	0.0055 ± 0.0073	0.801 *
uPAR	0.355, 0.270–0.549	0.324, 0.201–0.506	0.368, 0.229–0.437	0.268, 0.173–0.388	0.312
IL6/IL8	3.90, 1.89–5.82	3.85, 2.13–8.84	3.19, 1.96–5.05	2.41, 1.99–3.99	0.538

*: One Way Analysis of Variance (mean ± standard deviation). KW: Kruskal–Wallis One-Way Analysis of Variance on Ranks (median, 25–75%).

**Table 3 biomedicines-13-00531-t003:** Inflammatory biomarker levels in T2D subjects carrying different genotypes of telomerase gene (*TERC*, *TERT*, and *ACYP2*) polymorphisms (SNPs).

Genotype	IL6 Level	IL8 Level	TREM1 Level	uPAR Level	IL6/IL8 Ratio
Tested SNPs of TE genes *TERC rs12696304 G/C*					
GG	0.116, 0.090–0.215	0.024 ± 0.021	0.000 ± 0.007	0.296, 0.233–0.372	3.04, 2.00–4.67
GC	0.132, 0.103–0.174	0.033 ± 0.024	0.008 ± 0.014	0.339, 0.205–0.410	3.77, 2.29–7.40
CC	0.112, 0.094–0.159	0.039 ± 0.027	0.008 ± 0.009	0.315, 0.235–0.527	2.88, 1.62–5.32
*p*-value, KW	*p* = 0.463	*p* = 0.179 *	*p* = 0.060 *	*p* = 0.741	*p* = 0.473
*TERT rs2736100 C/A*					
CC	0.125, 0.100–0.180	0.051, 0.043–0.056	0.007 ± 0.000	0.369, 0.255–0.577	2.52, 1.67–4.73
CA	0.124, 0.093–0.183	0.033, 0.019–0.045	0.007 ± 0.013	0.321, 0.248–0.409	3.52, 2.20–4.88
AA	0.123, 0.092–0.160	0.027, 0.016–0.043	0.006 ± 0.010	0.296, 0.198–0.452	3.13, 1.65–7.08
*p*-value, KW	*p* = 0.974	*p* = 0.057	*p* = 0.552 *	*p* = 0.782	*p* = 0.880
*ACYP2 rs6713088 G/C*					
GG	0.124, 0.095–0.216	0.033 ± 0.019	0.007 ± 0.012	0.365, 0.294–0.579	3.64, 2.69–7.76
GC	0.123, 0.094–0.163	0.030 ± 0.023	0.005 ± 0.009	0.296, 0.202–0.394	3.15, 1.91–6.51
CC	0.126, 0.087–0.156	0.041 ± 0.029	0.007 ± 0.015	0.340, 0.237–0.476	2.88, 1.53–5.19
*p*-value, KW	*p* = 0.778	*p* = 0.200 *	*p* = 0.841 *	*p* = 0.210	*p* = 0.392

SNP: Single nucleotide polymorphism, *: One Way Analysis of Variance (mean ± standard deviation). KW: Kruskal–Wallis One-Way Analysis of Variance on Ranks (median, 25–75%).

## Data Availability

The datasets used and/or analyzed during the current study are available from the corresponding authors upon reasonable request.

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
