# Peer review of "A Perceived Dissociation Between Systemic Chronic Inflammation, Age, and the Telomere/Telomerase System in Type 2 Diabetes"

_biomedicines, 2025, doi:10.3390/biomedicines13030531_

Round 1
Reviewer 1 Report
Comments and Suggestions for Authors
Relevance and context:
The authors begin with a perfectly correct account of the common fact about the association of chronic inflammation and telomere leukocyte narrowing (TL), as well as the relationship of type 2 diabetes (DM) with TTL narrowing. They also indicate precisely that all three variables - inflammation, TL and DM - are closely related to the aging process. Against this background, trying to investigate what - inflammation, age or a combination of these - is at the root of telomere disorders in diabetic patients seems extremely logical and timely. Given the growing DM epidemic and population ageing, understanding the mechanisms linking these processes is crucial for developing preventive and therapeutic strategies. The relevance of the topic is beyond doubt.
Methodology of the study:
The chosen study scheme - cross-observational study - is quite adequate for initial examination of the claimed relationship. The study of 118 patients with DM is a representative sample for this type of studies, especially at the initial stage.
The methods used for analysis are also standard and reliable in this field: ELISA for measuring levels of inflammatory markers (IL-6, IL-8, TREM-1, uPAR) and telomerase in blood plasma. The choice of these markers of inflammation, although it requires additional justification in the full version of the article, is generally relevant, as they play an important role in inflammatory processes and metabolic disorders. However, it might be worth considering other key inflammatory mediators specific to CH2 or aging.
qPCR (quantitative polymerase real-time chain reaction) for VCT evaluation. PCR is the gold standard for measuring telomere length in population studies.
PCR for the genotyping of mononucleotide polymorphisms of telomerase genes (TERT, TERC, ACYP2). The study of genetic variants of telomerase genes is an important aspect, as genetics plays a significant role in the variability of TL and activity of the telomerase. These three genes are also chosen because of their key role in the telomerase complex.
Research results:
The main and most important result of the study presented is the absence of correlation between the levels of examined markers of inflammation and TL, the level of telomerase and age in patients with DM. There were also no significant differences in levels of inflammatory markers between groups of patients, divided into TL and telomerase levels, or between carriers of different genes of the telomere genes.
It is important to emphasize that negative results are as valuable in science as positive ones. In this case, the absence of expected correlation is an important result that indicates that the relationship between inflammation and telomeres in DM may not be as straightforward and simple as previously assumed. This does not devalue the study, but rather emphasizes the need for a deeper and more focused approach to the problem.
Interpretation and conclusion:
The authors' conclusion that no association was found between levels of tested markers of inflammation and TL, plasma telomerase level and age in DM is fully consistent with the results. The authors' suggestion of possible explanations for the dissociation with previously known associations in DM, although briefly mentioned in abstract, requires a detailed discussion in the full version of the article. It is essential that the authors analyze in detail the possible causes of the negative results obtained and propose alternative hypotheses or directions for further research.
To increase the likelihood of publication and enhance the scientific significance of the work, I would recommend to the authors:
Expand the discussion into a full article. It is necessary to discuss in detail possible reasons for the absence of expected correlations. Consider, in particular:
Choice of inflammation markers: Are the selected markers most relevant for studying the effect on telomeres in the context of DM? It may be worth considering other markers that reflect different aspects of the inflammatory process, such as chronic low-level inflammation markers, oxidative stress markers, or specific cytokinetic involved in DM pathogenesis.
Cross-sectional design: Does cross-sectional design limit the ability to identify dynamic relationships between inflammation and TL? Perhaps longitudinal studies tracking changes in time would be more informative.
Heterogenicity of DM: DM is a heterogeneous disease. It is possible that the relationship between inflammation and telomeres exists only in certain subgroups of patients with DM, for example depending on the stage of the disease, the presence of complications, or genetic factors. The article should discuss whether DM heterogeneity was considered in this study.
Telomerase measurement level: The measurement of plasma telomerase may not reflect the activity of telomerase in white cell cells, where telomeres are most relevant. Perhaps it is worth considering measuring telomerase activity directly in leukocytes.
Other factors affecting telomere length: In addition to inflammation, many other factors affect the length of telomere such as age, genetics, lifestyle (diet, physical activity, smoking, stress) etc. It is important to discuss in the article, how these factors could affect the results of the study and whether they were considered in the analysis.
Strengthen the "Introduction" section. To justify in more detail the choice of specifically investigated inflammatory markers and telomerase genes, based on literature data. Clearly formulate the hypothesis that the authors planned to test and explain why negative results are also important in the context of this hypothesis.
Finally, to emphasize the scientific value of negative results. Emphasize that the data obtained do not completely disprove the link between inflammation and telomeres in DM, but rather indicate the need to revise existing perceptions and search for more complex and nuanced mechanisms of interaction. Propose specific areas for further research based on the results and discussion.
Comments on the Quality of English Language
English in the manuscript is understood, easy to read and quite adequate
Author Response
We would like to kindly request the reviewer’s permission to quote some of their insightful comments in our revised manuscript, as we found them to be highly valuable. Thank you for the constructive feedback.
Com. The choice of these markers of inflammation, although it requires additional justification in the full version of the article, is generally relevant, as they play an important role in inflammatory processes and metabolic disorders. However, it might be worth considering other key inflammatory mediators specific to CH2 or aging.
Res. Agree, some explanation is added. Since we don’t know the nature of inflammation and its relevant markers in T2D, or in the telomere system, we selected a diverse type of biomarkers to broaden the scope and increase the chance of detecting any evidence of inflammation. The same markers were confirmed to be associated with T2D.
Com. It is worth to emphasize that negative results are as valuable as positive ones in this study, the absence of expected correlation is an important result that indicates that the relationship between inflammation and telomeres in DM may not be as straightforward and simple as previously assumed. This study, emphasizes the need for a deeper and more focused approach to the problem.
Res. Done. We have added this suggestion in the discussion section.
The authors' suggestion of possible explanations for the dissociation with previously known associations in DM, although briefly mentioned in abstract, requires a detailed discussion in the full version of the article. It is essential that the authors analyze in detail the possible causes of the negative results obtained and propose alternative hypotheses or directions for further research. Expand the discussion into a full article. It is necessary to discuss in detail possible reasons for the absence of expected correlations.
Res. Done. The explanation is explicitly discussed at the bottom of the discussion with a focus on 4 to 5 main points as suggested. Also, these points are addressed as limitations in the current study and were suggested for future studies. Furthermore, A short paragraph is added to the materials and methods section to justify the selection of markers.
Com. Choice of inflammation markers: Are the selected markers most relevant for studying the effect on telomeres in the context of DM?. Does cross-sectional design limit the ability to identify dynamic relationships between inflammation and TL? whether DM heterogeneity was considered in this study. The measurement of plasma telomerase may not reflect the activity of telomerase in white cell cells. Other factors affecting telomere length: In addition to inflammation, many other factors affect the length of telomere such as age, genetics, lifestyle.
Res. A paragraph is added to the discussion including the above-mentioned shortcoming. Thanks for the insightful suggestions.
Com. Strengthen the "Introduction" section. To justify in more detail the choice of specifically investigated inflammatory markers and telomerase genes, based on literature data. Clearly formulate the hypothesis that the authors planned to test and explain why negative results are also important in the context of this hypothesis.
Res. The justification for the use of the inflammatory biomarkers and TE SNPs, is added to the material and methods. We didn’t expect these results to add a hypothesis, on the contrary, we aimed to identify the cause of LTL shortening in T2D, and we suspect that the SCI or age is likely to be the cause. However, we touched on this hypothesis in the discussion.
Finally, to emphasize the scientific value of negative results. Emphasize that the data obtained do not completely disprove the link between inflammation and telomeres in DM, but rather indicate the need to revise existing perceptions and search for more complex and nuanced mechanisms of interaction. Propose specific areas for further research based on the results and discussion.
Res. Done. This concept idea is added after the addition of the limitations at the bottom of the discussion section. We highlighted the importance of the negative results and mentioned how these results triggered new research ideas. We quoted part of the above comment.
Reviewer 2 Report
Comments and Suggestions for Authors
This is a study entitled: A perceived dissociation between systemic chronic inflammation, age, and telomere/telomerase system in type 2 diabetes.
The objective is to find out the relationship of inflammation markers with telomere length, telomerase activity, and age in subjects with diabetes mellitus and a group of healthy controls.
As a flaw in the study design is that the age between the groups is different and this has a significant impact on telomere length, ... Age 56.5 (52.0 - 64.0) vs 51.0, (49.0 - 55.0) p= <0.001.
Given that telomere length is known to be associated with age and inflammatory factors, is it possible to include age and (e.g.) years of diabetes mellitus evolution as confounding variables in multivariate linear regression analysis to support telomere length as an independent association with inflammatory factors?
Further discussion is needed on the relevant finding of correlation between age and telomerase activity ... was a significant correlation between age and TE 189 level (CC 0.324, p 0.005).
Author Response
Com. As a flaw in the study design is that the age between the groups is different and this has a significant impact on telomere length, ... Age 56.5 (52.0 - 64.0) vs 51.0, (49.0 - 55.0) p= <0.001.
Res. True there is a significant difference in age between the cases and controls, as the other tested parameters in table 1, however, in this study there is no comparison of any of the tested biomarkers or the telomerase enzyme or LTL between the T2D and controls. That was done in our other articles. The controls used in this study confirm that inflammation in T2D is behind the LTL shortening, assuming that the controls have no inflammation.
Even, if we compare cases and controls, the younger group is the healthier control group, not the diabetic. Then the expected result is that the controls should have longer LT.
Com. Given that telomere length is known to be associated with age and inflammatory factors, is it possible to include age and (e.g.) years of diabetes mellitus evolution as confounding variables in multivariate linear regression analysis to support telomere length as an independent association with inflammatory factors?
Res. This is true, but unfortunately, we don’t have the data about the disease duration. The neutral effect of age in this study was demonstrated in the lack of correlation between age and LTL in T2D (Fig. 1 A), and in healthy subjects (Fig. 1 B). Moreover, multivariate analysis mostly shortens the list of significant differences or correlations rather than adding new significant confounders.
Com. Further discussion is needed on the relevant finding of correlation between age and telomerase activity... was a significant correlation between age and TE level (CC 0.324, p 0.005).
Res. Thank you for this good suggestion. A paragraph with 5 references (highlighted in light blue in the ref. list, 41 -45) is added at the bottom of the discussion exploring the association of age with TE in healthy subjects.
Round 2
Reviewer 2 Report
Comments and Suggestions for Authors
I thank you for your attention to the comments made.
It has been demonstrated that age and telomere length maintain a correlation in healthy subjects, evidently subjects with diabetes mellitus 2 will have an accelerated loss of telomeres, it is important to consider that the age factor influences not to analyze differences between groups, it is the effect that can present when it initiates a greater telomere shortening in different biological ages. That is why it is suggested that in the design of the study the control group should have a similar mean age to the study group. Evidently the years of evolution of the disease are another determining factor in the correlation, it is a unfortunate not to have this information.
The information added in the discussion now allows us to know the limitations of the present study and to establish with references possible scenarios of the expression of telomerase activity and its association with age in healthy subjects.